# Scaling Laws for Masked-Reconstruction Transformers on Single-Cell Transcriptomics

**Ihor Kendiukhov**  *kendiukhov@gmail.com*
*Department of Computer Science*
*University of Tübingen*

**Reviewed on OpenReview:** *https://openreview.net/forum?id=a8rUQqionr*

## Abstract

Neural scaling laws—power-law relationships between loss, model size, and data—have been extensively documented for language and vision transformers, yet their existence in single-cell genomics remains largely unexplored. We present the first systematic study of scaling behaviour for masked-reconstruction transformers trained on single-cell RNA sequencing (scRNA-seq) data. Using expression profiles from the CELLxGENE Census, we analyze a standardized sweep with 512 highly variable genes and 200,000 cells, alongside additional fixed-$V = 512$ analyses that probe data-size effects and loss interpretation. Across six model sizes spanning over five orders of magnitude in parameter count (533 to $1.0 \times 10^8$ parameters), we fit the parametric scaling law $L = aP^{-\alpha} + c$ to validation mean squared error (MSE). The standardized $V = 512$, $D = 200\text{k}$ sweep exhibits clear power-law scaling on validation loss, and held-out test evaluation follows the same qualitative trend. We additionally report a fixed-$V = 512$ data-size sweep, an empirical check of cross-gene residual heterogeneity, and a limited direct Gaussian-NLL calibration at small model sizes. Under a homoscedastic Gaussian approximation, the asymptotic floor in this setting corresponds to approximately 2.3 bits of *irreducible uncertainty remaining* per masked gene position, not to a universal biological constant. We discuss implications for the design of single-cell foundation models and outline the additional matched sweeps and likelihood-based objectives needed to turn this preliminary quantity into a rigorous transcriptomic entropy estimate.

## 1 Introduction

A central empirical finding in deep learning is that, under appropriate conditions, the loss of a neural network follows a power law as model size, dataset size, or training compute increases (Kaplan et al., 2020; Hestness et al., 2017). These *scaling laws* have proven remarkably consistent across modalities: they hold for autoregressive language models (Kaplan et al., 2020; Hoffmann et al., 2022), vision transformers (Zhai et al., 2022), and multimodal systems (Cherti et al., 2023). Beyond their theoretical interest, scaling laws have immediate practical value: they enable researchers to predict the performance of expensive large-scale training runs from cheap small-scale experiments, thereby guiding resource allocation and architectural choices.

Single-cell RNA sequencing (scRNA-seq) has become a cornerstone of modern biology, generating expression profiles for millions of individual cells across diverse tissues, species, and disease states (Regev et al., 2017). In response, the community has developed increasingly large *foundation models* for single-cell data, including scVI (Lopez et al., 2018), scGPT (Cui et al., 2024), Geneformer (Theodoris et al., 2023), scBERT (Yang et al., 2022), and scFoundation (Hao et al., 2024). These models are typically evaluated on downstream tasks such as cell-type annotation, batch correction, or perturbation prediction, but—in contrast to the language modelling literature—the fundamental question of how pretraining loss scales with model size and data volume has received almost no attention.

This gap matters for several reasons. First, without scaling laws, practitioners cannot determine whether a larger model will yield meaningful improvements or whether the current bottleneck is data, compute, or

intrinsic noise. Second, the asymptotic loss floor $c$ in a scaling law of the form $L = aP^{-\alpha} + c$ provides an empirical upper bound on the *reducible* error, which can be related to the intrinsic entropy of the data-generating process (Kaplan et al., 2020; Hoffmann et al., 2022). For transcriptomics, such an estimate would quantify the fundamental information content of gene expression measurements—a quantity of independent biological interest.

In this work, we take a first step towards establishing scaling laws for single-cell transcriptomics. We train masked-reconstruction transformers—a pretraining paradigm analogous to masked language modelling (Devlin et al., 2019)—on scRNA-seq data drawn from the CELLxGENE Census (CZI Single-Cell Biology, 2023), varying the model size across six presets that span from 533 to $1.0 \times 10^8$ trainable parameters. Our main experiment is a standardized $V = 512$, $D = 200k$ sweep, which we complement with additional fixed-$V = 512$ analyses across dataset size and loss diagnostics.

**Contributions.**

1. We present, to our knowledge, the first systematic investigation of neural scaling laws for masked-reconstruction pretraining on scRNA-seq data.

2. We demonstrate that power-law scaling emerges in a standardized $V = 512$, $D = 200k$ sweep, with held-out test evaluation following the same qualitative ordering across model sizes.

3. We add additional fixed-$V = 512$ analyses, including a data-size sweep, a representative cross-gene residual heterogeneity diagnostic, and a small-scale direct Gaussian-NLL calibration.

4. We provide a preliminary estimate of the asymptotic uncertainty floor for masked reconstruction in this setting and state the assumptions required to interpret it in information-theoretic units.

5. We identify the key experimental ingredients—matched $V/D$ sweeps, likelihood-based objectives, and comprehensive compute accounting—needed to progress from loss-vs-parameters curves to transcriptomic entropy estimates.

## 2   Related Work

**Neural scaling laws.**   Hestness et al. (2017) first documented power-law scaling of test error with dataset size across several domains. Kaplan et al. (2020) established that language model cross-entropy loss follows $L(P) = (P_c/P)^{\alpha_P}$ over several orders of magnitude in parameter count $P$, with $\alpha_P \approx 0.076$ for autoregressive transformers. Importantly, they found that scaling with model size, data size, and compute each follows its own power law, and that training can be either model-limited or data-limited depending on the regime. Hoffmann et al. (2022) refined these findings, demonstrating that Kaplan *et al.*'s protocol over-allocates parameters relative to data and proposing compute-optimal (*Chinchilla*) scaling rules. Subsequent work has extended scaling laws to vision transformers (Zhai et al., 2022), contrastive multimodal models (Cherti et al., 2023), and various other architectures and tasks (Clark et al., 2022; Muennighoff et al., 2024).

**Single-cell foundation models.**   The single-cell community has developed several large pretrained models. scVI (Lopez et al., 2018) introduced variational autoencoders for scRNA-seq, and subsequent work extended this framework to multi-omic data (Gayoso et al., 2022). More recently, transformer-based architectures have gained prominence: scGPT (Cui et al., 2024) adopts a generative pretraining approach with gene tokens; Geneformer (Theodoris et al., 2023) pretrains on rank-ordered gene expression using a BERT-like masked objective; scBERT (Yang et al., 2022) similarly uses masked gene expression prediction; and scFoundation (Hao et al., 2024) scales to 100 million parameters trained on 50 million cells. While these works report downstream benchmark results, none systematically characterizes how pretraining loss scales with model size—the question we address here.

**Masked autoencoders for genomics.**   Masked reconstruction is a well-established self-supervised pre-training strategy. In natural language processing, BERT (Devlin et al., 2019) popularized masked token prediction. In computer vision, masked autoencoders (MAE) (He et al., 2022) demonstrated that masking

high fractions of image patches leads to effective visual representations. For single-cell data, the masked reconstruction objective is a natural fit: by randomly masking a subset of gene expression values and training the model to reconstruct them from the remaining genes, the model learns co-expression structure and gene regulatory relationships. Our architecture follows this paradigm, using a permutation-invariant transformer encoder with a per-gene reconstruction head.

## 3 Data

### 3.1 Source, Preprocessing, and Dataset Variants

All data are drawn from the CELLxGENE Census (CZI Single-Cell Biology, 2023), a curated aggregation of publicly available single-cell RNA-seq datasets encompassing diverse human tissues and cell types. The main experiment uses 200,000 cells with 512 highly variable genes (HVGs), stored as `census_human.D200000.V512.log1p.h5ad`. For additional fixed-vocabulary analyses, we construct matched-preprocessing subsets with the same $V = 512$ gene set and $D \in \{25k, 50k, 100k, 200k\}$ cells.

The preprocessing pipeline proceeds as follows. First, we load the expression matrix from the Census and select the expression layer with a valid library size (preferring raw counts where available). Zero-library cells are removed. Second, we select highly variable genes using the Seurat v3 method (Stuart et al., 2019), which ranks genes by their variance-to-mean ratio after variance-stabilizing transformation. Third, we apply library-size normalization to a target sum of $10^4$ followed by the $\log(1 + x)$ transform. The resulting matrix is stored in float32. Finally, we split cells into training (90%), validation (5%), and test (5%) partitions using stratified sampling based on available cell-type annotations where possible, with random splitting as fallback.

## 4 Model Architecture

### 4.1 Overview

We employ a permutation-invariant transformer encoder designed for set-structured inputs, where each element of the set corresponds to a gene. The architecture follows the standard transformer encoder (Vaswani et al., 2017) but omits positional encodings entirely, since the ordering of genes carries no inherent biological meaning. The model receives three inputs: gene identifiers (integer indices into a learned embedding table), observed expression values, and a binary mask indicating which values have been hidden.

### 4.2 Input Embedding

Each gene token is formed by summing two components:

1. A **gene identity embedding** $\mathbf{e}_g \in \mathbb{R}^d$, obtained by looking up the gene index in a learned embedding table $\mathbf{E} \in \mathbb{R}^{V \times d}$.

2. A **value projection** $\mathbf{v}_g = \mathbf{W}_v x_g + \mathbf{b}_v$, where $\mathbf{W}_v \in \mathbb{R}^{d \times 1}$ and $x_g$ is the (possibly masked) expression value for gene $g$.

At masked positions, the value projection is replaced by a learned **mask token** $\mathbf{m} \in \mathbb{R}^d$, initialized to zero and updated during training. The resulting token representation for gene $g$ is:

$$\mathbf{h}_g^{(0)} = \mathbf{e}_g + \begin{cases} \mathbf{v}_g & \text{if gene } g \text{ is observed} \\ \mathbf{m} & \text{if gene } g \text{ is masked.} \end{cases} \tag{1}$$

### 4.3 Transformer Encoder

The token representations $\{\mathbf{h}_g^{(0)}\}_{g=1}^V$ are processed by a stack of $L$ standard transformer encoder layers (Vaswani et al., 2017), each comprising multi-head self-attention and a position-wise feed-forward network

Table 1: Model size presets used in the standardized $V = 512$ experiments. $d$: model dimension; $L$: number of encoder layers; $H$: number of attention heads; FFN mult: feed-forward expansion factor.

| Size | $d$ | $L$ | $H$ | FFN mult | Params ($V = 512$) |
|------|-----|-----|-----|----------|--------------------|
| XXS  | 1   | 1   | 1   | 1        | 533                |
| TINY | 16  | 1   | 1   | 1        | 9,953              |
| XS   | 64  | 2   | 4   | 4        | 132,993            |
| S    | 128 | 4   | 8   | 4        | 859,137            |
| M    | 512 | 6   | 8   | 4        | 19,178,497         |
| L    | 1020| 8   | 12  | 4        | 100,510,801        |

with GELU activation (Hendrycks and Gimpel, 2016). We use pre-layer normalization (Pre-LN) (Xiong et al., 2020), which places layer normalization before (rather than after) the attention and feed-forward sub-layers, yielding more stable training dynamics at scale. The feed-forward hidden dimension is $4d$ by default.

### 4.4 Prediction Head and Pooling

The output of the encoder at each gene position is passed through a linear prediction head that maps $\mathbb{R}^d \to \mathbb{R}^1$, producing a scalar expression estimate $\hat{x}_g$ for each gene. The loss is computed only at masked positions (Section 5). For downstream applications requiring a cell-level representation, the model produces a pooled embedding $\mathbf{z} \in \mathbb{R}^d$ via mean-pooling over all gene tokens.

### 4.5 Model Sizes

To study scaling behaviour, we define six size presets that span approximately five orders of magnitude in parameter count. Table 1 summarizes their configurations for the standardized $V = 512$ setting studied in the main text.

## 5 Training

### 5.1 Masked Reconstruction Objective

Following the masked language modelling paradigm (Devlin et al., 2019), we randomly mask 15% of gene expression values in each training sample. The training objective is to minimize the mean squared error (MSE) between the model's predictions and the true expression values at masked positions only:

$$\mathcal{L} = \frac{1}{|\mathcal{M}|} \sum_{g \in \mathcal{M}} \left( \hat{x}_g - x_g \right)^2, \tag{2}$$

where $\mathcal{M}$ denotes the set of masked gene indices and $|\mathcal{M}|$ its cardinality. We also monitor masked mean absolute error (MAE) as a secondary metric.

### 5.2 Optimization

We use AdamW (Loshchilov and Hutter, 2019) with weight decay $\lambda = 0.01$ and gradient clipping at norm 1.0. For the main GPU runs, the base learning rate is $3.125 \times 10^{-5}$, computed as $\mathrm{lr_{base}} \cdot \mathrm{eff\_batch}/256$ where $\mathrm{lr_{base}} = 2.5 \times 10^{-4}$ and the effective batch size is 32 (physical batch of 4 with 8 gradient accumulation steps). All GPU runs use mixed-precision training (float16 autocast) for efficiency. The standardized parameter sweep, the fixed-$V = 512$ data-size sweep, and the direct-NLL calibration reruns all use 60,000 optimization steps.

Table 2: Standardized $V=512$ summary (200k cells, 60k steps, 3 seeds per size). Mean $\pm$ standard deviation is reported across seeds for both validation and held-out test MSE.

| Size | Params | Val. MSE | Test MSE |
|------|--------|----------|----------|
| XXS | 533 | $2.101 \pm 0.032$ | $2.110 \pm 0.033$ |
| TINY | 9,953 | $1.600 \pm 0.010$ | $1.617 \pm 0.007$ |
| XS | 132,993 | $1.515 \pm 0.011$ | $1.533 \pm 0.007$ |
| S | 859,137 | $1.482 \pm 0.008$ | $1.500 \pm 0.007$ |
| M | 19,178,497 | $1.462 \pm 0.003$ | $1.479 \pm 0.004$ |
| L | 100,510,801 | $1.477 \pm 0.009$ | $1.494 \pm 0.011$ |

### 5.3 Evaluation Protocol

We evaluate on the held-out validation split every 1,000 steps, saving the checkpoint with the lowest validation MSE (`ckpt_best.pt`). The final scaling analysis uses the best validation MSE achieved during training for each run, which we denote $L^*$. Each model size in the standardized sweep is trained with three random seeds (7, 8, 9) to characterize run-to-run variance.

## 6 Scaling Law Formulation

Following Kaplan et al. (2020), we model the relationship between validation MSE and parameter count $P$ as a power law with an additive offset:

$$L(P) = a \cdot P^{-\alpha} + c, \tag{3}$$

where $\alpha > 0$ is the *scaling exponent* governing the rate of improvement, $a > 0$ is a multiplicative constant, and $c \geq 0$ represents the *irreducible loss floor*—the error that cannot be reduced by increasing model capacity, attributable to intrinsic data noise, biological stochasticity, or information lost during preprocessing.

We fit Equation (3) by grid search over candidate values of $c$ in $[0, 0.99 \cdot \min_i L_i]$, and for each candidate perform ordinary least squares on the linearized form $\log(L - c) = \log a - \alpha \log P$. The fit with the highest $R^2$ on the log-scale residuals is selected. All main-text fits are computed only on standardized sweeps with matched hyperparameters.

## 7 Results

### 7.1 Overview of Experiments

Our main quantitative evidence is a standardized $V = 512$, $D = 200k$ sweep: 18 runs (6 model sizes $\times$ 3 seeds), all trained for 60,000 steps with identical optimization hyperparameters on the same dataset. We complement this sweep with two additional analyses drawn from the same preprocessing pipeline: a fixed-$V = 512$ data-size sweep using the M model at $D \in \{25k, 50k, 100k, 200k\}$, and diagnostic analyses of loss interpretation including a representative residual-heterogeneity check and small direct-NLL reruns for XXS and TINY.

In the standardized $V = 512$ sweep, MSE decreases steadily from XXS through M on both validation and held-out test data. The L model does not improve further and is slightly worse than M on average, indicating that the loss curve has already begun to flatten within the explored parameter range.

### 7.2 Main Parameter-Scaling Results

Table 3 reports the power-law fit parameters for the standardized $V = 512$ sweep on both validation and held-out test MSE, and Figure 1 shows the corresponding log-log fits.

The standardized $V = 512$ regime with 200,000 cells exhibits a clear scaling relationship. On validation data, the fitted exponent is $\alpha = 0.266$ with $R^2 = 0.858$. On held-out test data, the fit is nearly identical

Table 3: Power-law fits for the standardized $V=512$ sweep (200k cells, 18 runs). Reporting both validation and held-out test fits confirms that the qualitative conclusion does not depend on checkpoint selection by validation alone.

| Split | $n$ | $\alpha$ | $a$ | $c$ | $R^2$ |
|---|---|---|---|---|---|
| Validation | 18 | 0.266 | 2.153 | 1.444 | 0.858 |
| Test | 18 | 0.268 | 2.168 | 1.462 | 0.859 |

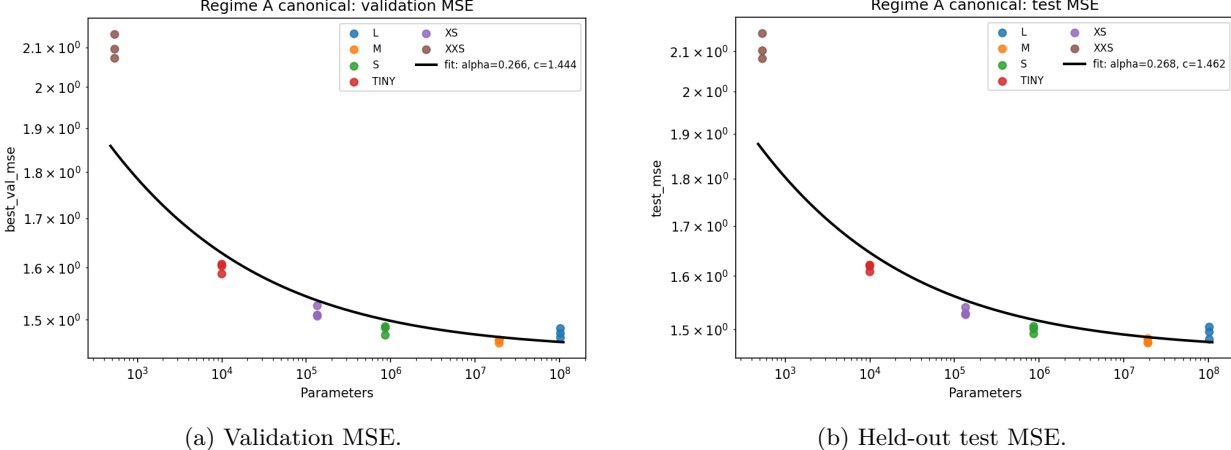

(a) Validation MSE.              (b) Held-out test MSE.

Figure 1: Standardized $V=512$ scaling on log-log axes. Each point is one training run (best checkpoint per seed). The held-out test fit closely matches the validation fit, with $\alpha \approx 0.267$ and $R^2 \approx 0.859$ on both splits.

($\alpha = 0.268$, $R^2 = 0.859$), showing that the trend is not merely an artifact of selecting checkpoints by validation performance. In the standardized fit, each tenfold increase in parameter count reduces the excess loss ($L - c$) by a factor of approximately $10^{0.266} \approx 1.84$. The irreducible floor is also stable across splits: $c \approx 1.44$ on validation and $c \approx 1.46$ on test.

The observation that L-size models ($\sim$101M parameters) do not improve over M-size models ($\sim$19M parameters) is consistent with the fitted curve, which has largely flattened by $P \approx 10^7$. This provides practical guidance: for this dataset and task, models beyond the tens-of-millions scale offer diminishing returns.

### 7.3 Additional Analyses at Fixed $V=512$

To complement the parameter-scaling results, we analyzed two additional $V = 512$ experiments from the same preprocessing pipeline. First, we evaluated a fixed-model data-size sweep using the M model at $D \in \{25k, 50k, 100k, 200k\}$ with 3 seeds per point and the same 60k-step training budget. Second, we computed a per-gene held-out residual diagnostic on a representative L run to test the homoscedasticity assumption underlying the MSE-to-entropy conversion.

The fixed-$V$ data-size sweep yields a nearly flat validation fit, $\beta = -0.0023$ with $R^2 = 0.611$, rather than a clean monotonic decrease in MSE with larger $D$. This does not imply that dataset size is unimportant; rather, it shows that at fixed total steps, larger datasets are seen for fewer effective epochs, so the current sweep is not yet a definitive $L(D)$ estimate.

The residual diagnostic also shows that the homoscedastic Gaussian approximation is imperfect. In the representative held-out L run, the across-gene coefficient of variation of per-gene test MSE is 0.730, the ratio between the 90th and 10th percentiles is 10.1, and the correlation between mean expression and per-gene MSE is 0.892. We therefore treat the entropy-style conversion from MSE to bits as a heuristic estimate of irreducible uncertainty under the task and preprocessing pipeline, not as a justified gene-wise generative model.

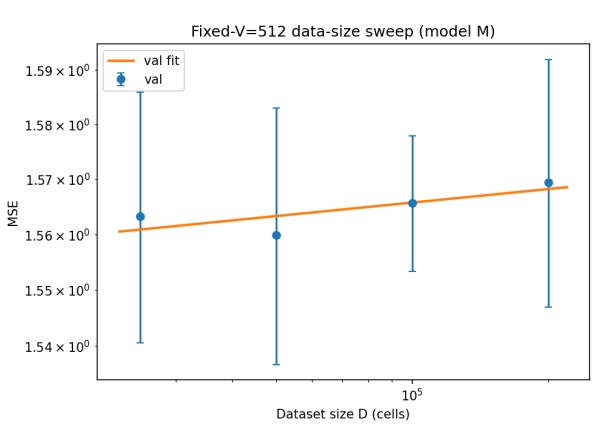
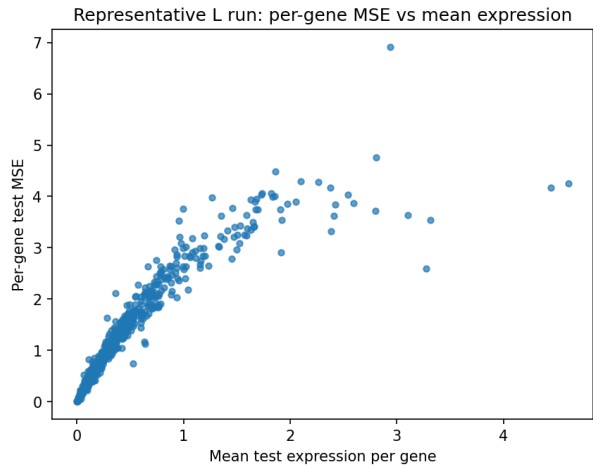

(a) Fixed-$V$ data-size sweep at model size M.

(b) Per-gene test MSE vs. mean expression.

Figure 2: Additional analyses at $V\!=\!512$. Left: at fixed 60k steps, the M-model data-size sweep is nearly flat in validation MSE ($\beta \approx -0.002$, $R^2 = 0.61$), indicating that a clean $L(D)$ law still requires matched optimization budgets. Right: residual error is strongly gene-dependent in a representative held-out L run, with correlation 0.892 between mean expression and per-gene MSE.

### 7.4 Comparison with Language Model Scaling

The scaling exponent $\alpha \approx 0.266$–$0.268$ observed in the standardized $V\!=\!512$ sweep is substantially larger than the $\alpha_P \approx 0.076$ reported by Kaplan et al. (2020) for autoregressive language models, or the $\alpha_P \approx 0.34$ reported by Hoffmann et al. (2022) after correcting for compute-optimal training. Direct comparison is complicated by several factors: we use MSE rather than cross-entropy (Appendix A), our masking rate (15%) is much lower than typical language model sequence lengths, and the structure of gene expression data differs fundamentally from natural language. Nevertheless, the qualitative finding—that loss follows a power law with diminishing returns at large scale—is consistent across domains.

As in prior scaling-law work, different empirical axes need not follow identical exponents. In our setting, the current fixed-$V\!=\!512$ data-size analysis is too limited to support a full Section 6.3-style intersection analysis in the main text.

### 7.5 Preliminary Entropy Estimate for the Main $V\!=\!512$ Sweep

A central motivation for fitting scaling laws with an additive offset $c$ is that the irreducible floor can, under appropriate assumptions, be converted to an estimate of the *intrinsic entropy* of the data-generating process—the information content per prediction that no model can recover. While a full entropy estimate requires matched data/compute sweeps and a likelihood-based objective (Section 9), we present a preliminary estimate for the standardized $V\!=\!512$ sweep using Gaussian approximations to the fitted floor values.

#### 7.5.1 Gaussian NLL Derivation from MSE

Our standardized training runs record MSE as the primary loss. To obtain a Gaussian negative log-likelihood (NLL), we derive it post hoc under the assumption that the residual errors at masked positions are approximately Gaussian with variance equal to the MSE and, crucially, that this residual variance can be treated as homoscedastic across genes. Specifically, for a Gaussian with variance $\sigma^2$, the NLL per position is:

$$\mathcal{L}_{\text{NLL}} = \tfrac{1}{2}\ln(2\pi\sigma^2) + \tfrac{1}{2}, \tag{4}$$

and the differential entropy (in bits) is:

$$h = \tfrac{1}{2}\log_2(2\pi e\,\sigma^2). \tag{5}$$

Table 4: Scaling-law fits for the standardized $V = 512$ sweep (200k cells, 18 runs), with entropy floor estimates. The NLL values are derived from MSE under a Gaussian assumption (Equation 4), so the two estimates are not fully independent; Figure 2 shows that residual variance is in fact gene-dependent.

| Metric | $n$ | $\alpha$ | $a$ | $c$ | $R^2$ | Entropy floor (bits/masked pos.) |
|---|---|---|---|---|---|---|
| MSE | 18 | 0.266 | 2.153 | 1.444 | 0.858 | 2.312 |
| Gauss. NLL | 18 | 0.182 | 0.401 | 1.592 | 0.838 | 2.296 |

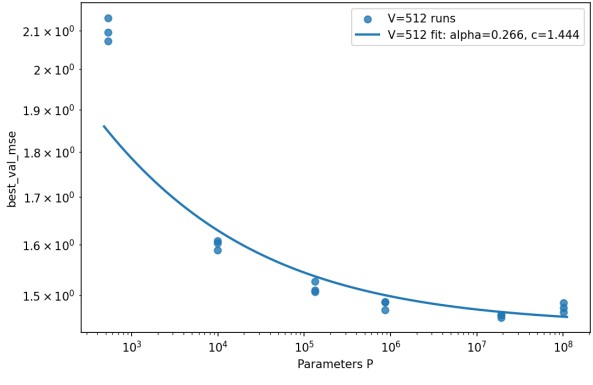 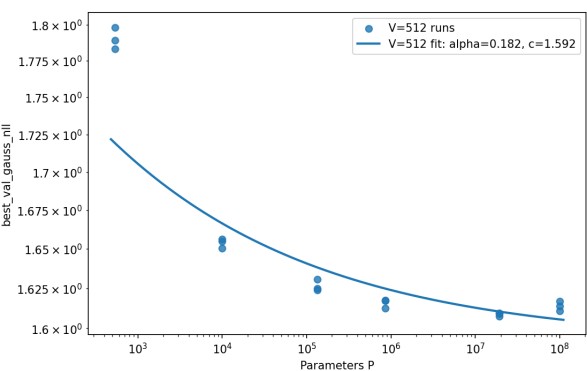

(a) MSE vs. parameters ($\alpha = 0.266$, $c = 1.444$).  (b) Gaussian NLL vs. parameters ($\alpha = 0.182$, $c = 1.592$).

Figure 3: Standardized $V = 512$ scaling fits (18 runs, 6 sizes $\times$ 3 seeds, all 60k steps). Left: MSE metric. Right: derived Gaussian NLL metric. Under the Gaussian homoscedastic approximation, both correspond to about 2.30 bits of irreducible uncertainty remaining per masked position.

Using a standardized set of 18 runs (6 sizes $\times$ 3 seeds, all trained for 60,000 steps with identical hyperparameters), we refit the scaling law for both the MSE and the derived Gaussian NLL metrics. As a limited direct check on the conversion, we also reran XXS and TINY under the same $V = 512$, $D = 200$k, 60k-step protocol with validation Gaussian NLL logged directly; those results are used only as a calibration check, not as a replacement for the six-size standardized fit. Table 4 reports the results.

Figure 3 shows the scaling curves for both metrics on the standardized run set.

### 7.5.2 Entropy Estimate

The two derivations yield closely agreeing estimates:

- **From MSE floor** ($c_{\text{MSE}} = 1.444$): interpreting $c$ as the irreducible variance $\sigma^2$ and applying Equation (5) gives $h = \frac{1}{2} \log_2(2\pi e \cdot 1.444) = 2.312$ bits per masked position.

- **From NLL floor** ($c_{\text{NLL}} = 1.592$ nats): converting to bits via $h = c_{\text{NLL}} / \ln 2 = 2.296$ bits per masked position.

Both approaches converge on an estimate of approximately **2.30 bits of irreducible uncertainty remaining per masked gene position** for the $V = 512$ regime, but only under the stated Gaussian and homoscedasticity assumptions.

### 7.5.3 Caveats

This estimate is preliminary and subject to several important caveats. First, the Gaussian NLL values in the standardized sweep are *derived from* the MSE values (not independently logged during training), so the two estimates in Table 4 are not fully independent—their agreement primarily reflects the self-consistency of the

Gaussian assumption rather than an independent validation. The direct-logging check partially addresses this only at small scale: across three-seed reruns, the mean validation Gaussian NLL differs from the MSE-derived value by 0.022 nats for XXS (1.768 direct vs. 1.790 derived) and 0.020 nats for TINY (1.634 direct vs. 1.654 derived). This supports the local calibration of Equation (7) for those two small models, but it does not replace the six-size standardized table, because the corresponding larger models were not trained under the same direct-NLL protocol.

Second, the entropy floor depends on the preprocessing pipeline. The $\log(1 + x)$ transform and library-size normalization change the scale and distribution of gene expression values; the 2.30-bit figure applies specifically to masked positions in $\log(1 + x)$-transformed, library-size-normalized expression over 512 HVGs. Changing the vocabulary size, normalization target, or transformation would alter the floor.

Third, a complete entropy estimate requires not only the $L(P)$ scaling law (loss vs. parameters) but also the $L(D)$ law (loss vs. dataset size) and ideally the intersection $L(C_{\min})$ of the compute-optimal frontier, following the methodology of Hoffmann et al. (2022). We include a fixed-$V = 512$ data-size sweep, but at a fixed 60k-step budget it is nearly flat in MSE and therefore does not yet provide a clean $L(D)$ law; a definitive estimate still requires matched optimization budgets across $D$.

Fourth, the homoscedasticity assumption is only approximate in our data. A representative held-out L run shows substantial cross-gene heterogeneity in residual error (coefficient of variation 0.730, $q_{90}/q_{10} = 10.1$, correlation 0.892 between mean expression and per-gene MSE; Figure 2). This does not invalidate the floor-fit itself, but it means the conversion from MSE to bits should be interpreted as a task-level approximation rather than a justified per-gene likelihood model.

Despite these limitations, the convergence of the two derivations on $\sim$2.30 bits provides a useful first benchmark: under the stated approximation, it suggests that roughly 2.3 bits of *irreducible uncertainty remain* per masked gene among the top 512 HVGs, given this data source, preprocessing pipeline, and masking task.

## 8 Discussion

### 8.1 Implications for Single-Cell Foundation Models

Our results carry several implications for the design and training of single-cell foundation models.

**Data-size effects depend on optimization budget.** The fixed-$V = 512$ data-size sweep shows that loss does not decrease monotonically with larger $D$ when total training steps are held fixed. This does not imply that dataset size is unimportant; rather, it shows that data volume, effective epochs, and optimization budget interact. The current evidence therefore supports a narrower conclusion: in the standardized $V = 512$, $D = 200$k setting, larger models help, while a clean data-law analysis still requires matched training budgets across dataset sizes.

**Diminishing returns beyond $\sim$20M parameters.** Within the standardized $V = 512$ sweep, the loss curve flattens around $10^7$ parameters. While this specific threshold depends on the dataset, gene vocabulary, and preprocessing choices, it suggests that for datasets of $\sim$200,000 cells, models in the tens of millions of parameters are sufficient to capture the learnable structure. This is a useful calibration point for practitioners deciding whether to invest in training larger models.

**The irreducible floor as a biological signal.** The estimated $c \approx 1.44$ (MSE) for the standardized $V = 512$ sweep, corresponding under the Gaussian homoscedastic approximation to approximately 2.30 bits of irreducible uncertainty remaining per masked gene position (Section 7.5), reflects the combined effect of biological noise, technical noise, and information loss from dimensionality reduction. Disentangling these contributions is an important direction for future work: if the floor is dominated by technical noise, it may be reducible through improved experimental protocols; if it reflects intrinsic biological stochasticity, it represents a fundamental limit on expression prediction. The 2.30-bit figure should therefore be interpreted as an

approximate, task-specific quantity of irreducible uncertainty under this preprocessing pipeline, masking objective, and gene vocabulary, rather than as a universal constant of transcriptomics.

## 8.2 Data, Compute, and Parameter Count

A plausible interpretation of our results is that scaling behaviour depends on the interaction between data volume, parameter count, and optimization budget. In the standardized sweep, even the L model ($\sim$101M parameters) is trained on roughly $200{,}000 \times 512 \approx 10^8$ scalar expression values. However, the fixed-$V = 512$ data-size analysis shows that holding steps fixed while varying $D$ is not enough to produce a clean data law, because the effective number of epochs changes with dataset size. The transition from effective to ineffective scaling therefore cannot be localized from the current matrix alone; isolating it requires matched $V/D$ sweeps with standardized optimization budgets.

We note that the effective data budget is further multiplied by the number of training steps and the mask rate: each cell is seen multiple times with different random masks, effectively augmenting the dataset. A full accounting of *tokens seen* (steps $\times$ effective batch size $\times V$) would enable a cleaner analysis of data-vs-compute scaling, which we leave to future work.

## 8.3 Comparison to Downstream Evaluations

Prior single-cell foundation models are typically evaluated on downstream tasks (cell-type classification, batch correction, perturbation response prediction) rather than pretraining loss. Our work is complementary: we focus on the pretraining loss itself, which is the quantity most directly comparable across model sizes and most amenable to scaling-law analysis. A natural extension would be to study whether downstream task performance also follows predictable scaling curves, as has been observed in some NLP settings (Wei et al., 2022; Isik et al., 2024).

# 9 Limitations and Future Work

**Incomplete deconfounding of data size and vocabulary size.** This paper establishes parameter scaling in one standardized $V = 512$, $D = 200k$ setting and adds a fixed-$V = 512$ data-size analysis, but it does not yet contain a full matrix varying $V$ and $D$ independently. Future work should construct matched datasets at multiple vocabulary sizes and cell counts so that $V$ and $D$ can be varied independently under a common preprocessing protocol.

**MSE as loss metric and derived entropy.** We use MSE as both the training loss and the scaling-law target. While we provide a preliminary entropy estimate of $\sim$2.30 bits per masked position for the standardized $V = 512$ sweep (Section 7.5), this estimate relies on a Gaussian assumption applied post hoc to MSE values, rather than a natively trained information-theoretic loss. The Gaussian NLL values in the standardized 18-run sweep are derived from MSE, not independently logged, so the two entropy derivations are not fully independent. To partially test this approximation, we completed direct-logging XXS/TINY reruns under the same $V = 512$, $D = 200k$, 60k-step setup; the mean validation Gaussian NLL is within about 0.02 nats of the value derived from best validation MSE for both sizes. However, the standardized scaling table itself remains MSE-based, because we do not yet have a fully matched six-size direct-NLL rerun under the same sweep configuration, and the homoscedastic Gaussian conversion remains only approximate. To obtain a rigorous transcriptomic entropy estimate, future work should train with a parametric Gaussian NLL that predicts both $\mu$ and gene-specific $\sigma^2$, or with a discretized cross-entropy over binned expression levels. Additionally, the full $L(D)$ and $L(C_{\min})$ intersection experiment described by Hoffmann et al. (2022) is needed to disentangle data-limited from model-limited floors.

**Limited model size range.** While our six presets span five orders of magnitude, the "useful" range for the standardized sweep is narrower: the XXS model (533 parameters) is too small to be informative about neural scaling behaviour, and models beyond L were not included in the standardized sweep. A denser sweep in the $10^5$–$10^7$ range, with more seeds, would yield tighter confidence intervals on $\alpha$ and $c$.

**Training compute as a scaling axis.** We have studied scaling with respect to parameters only. A complete picture requires also fitting $L(C)$ as a function of compute $C$ (measured in FLOPs or tokens processed), and $L(D)$ as a function of dataset size. Our runs do record tokens processed (steps $\times$ effective batch size $\times V$), but the additional analyses reported here do not yet replace a fully matched $V/D/C$ experimental matrix. Such a matrix remains necessary for a definitive Section 6.3-style entropy estimate.

**Generalization to other tissues, species, and modalities.** Our data are drawn from a single Census subset. It remains to be seen whether the observed scaling exponents and floors generalize across tissue types, developmental stages, species, or multi-omic modalities (ATAC-seq, spatial transcriptomics).

## 10 Broader Impact

This work is primarily methodological, but its conclusions inherit biases from the underlying single-cell compendium and from preprocessing choices such as HVG selection, library-size normalization, and the $\log(1+x)$ transform. Dataset composition can over-represent certain tissues, platforms, and donor populations, so any observed "irreducible" floor may partly reflect those biases rather than purely biological uncertainty. Similarly, the entropy-style quantity discussed in Section 7.5 is task-dependent: it is specific to masked reconstruction over a chosen gene vocabulary after a particular preprocessing pipeline, and it should not be interpreted as a universal constant of transcriptomics.

## 11 Conclusion

We have presented the first systematic study of neural scaling laws for masked-reconstruction transformers on single-cell RNA-seq data. Our principal finding is that scaling behaviour analogous to that observed in language modelling does emerge in masked-reconstruction single-cell pretraining under a standardized $V = 512$, $D = 200k$ setup. In that setting, validation MSE follows a power law in parameter count with an identifiable irreducible floor, and held-out test evaluation follows the same qualitative pattern. The fixed-$V = 512$ data-size sweep and residual diagnostics further show that data volume, optimization budget, and loss interpretation interact in ways that must be controlled explicitly for stronger causal or information-theoretic claims. A preliminary conversion of the asymptotic loss floor to information-theoretic units yields an estimate of approximately 2.30 bits of irreducible uncertainty per masked gene position under the stated Gaussian approximation.

These results establish a quantitative framework for reasoning about the scaling behaviour of single-cell foundation models. They suggest that the field's current emphasis on increasing model size should be balanced by equivalent investment in curating larger, more diverse training corpora and in running matched sweeps that disentangle dataset size, vocabulary size, and compute. With the addition of natively trained information-theoretic loss functions, matched experimental conditions, and comprehensive compute accounting, this framework can be refined to produce rigorous estimates of the intrinsic entropy of the transcriptome—a quantity of fundamental biological and information-theoretic interest.

## Acknowledgements

We thank the CELLxGENE team at the Chan Zuckerberg Initiative for maintaining the Census data resource, and the developers of scvi-tools, Scanpy, and PyTorch for the software infrastructure underlying this work. Compute was provided by consumer GPU hardware.

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

## A  On the Choice of Loss Metric

Mean squared error is the most common regression loss for continuous expression values, but it is not directly comparable to the cross-entropy losses used in language model scaling laws. Ideally, one would train with a Gaussian negative log-likelihood where the model predicts both a mean $\mu_g$ and a variance $\sigma_g^2$:

$$\mathcal{L}_{\text{NLL}} = \frac{1}{|\mathcal{M}|} \sum_{g \in \mathcal{M}} \left[ \frac{(x_g - \mu_g)^2}{2\sigma_g^2} + \log \sigma_g + \frac{1}{2} \log(2\pi) \right]. \tag{6}$$

This formulation measures loss in nats (convertible to bits by dividing by $\ln 2$) and would allow the asymptotic floor $c$ to be directly interpreted as an entropy estimate.

In the current study, we derive Gaussian NLL values post hoc from MSE by assuming homoscedastic Gaussian residuals with variance $\sigma^2 = \text{MSE}$:

$$\hat{\mathcal{L}}_{\text{NLL}} = \tfrac{1}{2} \ln(2\pi \cdot \text{MSE}) + \tfrac{1}{2}. \tag{7}$$

This derivation is exact only if the prediction errors are truly $\mathcal{N}(0, \text{MSE})$ with a variance that can be treated as constant across genes, but it conflates aleatoric and epistemic uncertainty and cannot capture heteroscedastic noise across genes. In our standardized 18-run sweep, all Gaussian NLL values carry the provenance label `derived_from_best_mse` in the analysis pipeline, confirming that they are not independently measured. As a partial empirical check, we reran XXS and TINY under the same $V = 512$, $D = 200k$, 60k-step configuration with validation Gaussian NLL logged directly. Across three seeds, the directly logged validation Gaussian NLL is 1.768 vs. 1.790 derived for XXS, and 1.634 vs. 1.654 derived for TINY, *i.e.* within about 0.02 nats in both cases. These small-model reruns support the local numerical calibration of the conversion, but they do not constitute a full six-size direct-NLL replacement for the standardized table because the corresponding larger models were not trained under the same direct-NLL protocol. A held-out per-gene residual diagnostic on a representative L run gives a test-MSE coefficient of variation of 0.730 and a $q_{90}/q_{10}$ ratio of 10.1, reinforcing that the homoscedastic conversion is only approximate.

The entropy conversion from each floor then proceeds as follows:

$$h_{\text{NLL}} = c_{\text{NLL}}/\ln 2 = 1.592/0.693 = 2.296 \text{ bits,} \tag{8}$$

$$h_{\text{MSE}} = \tfrac{1}{2} \log_2(2\pi e \cdot c_{\text{MSE}}) = \tfrac{1}{2} \log_2(2\pi e \cdot 1.444) = 2.312 \text{ bits.} \tag{9}$$

The close agreement ($\Delta = 0.016$ bits) reflects the self-consistency of the Gaussian assumption rather than independent validation. A per-gene residual diagnostic therefore matters for interpretation: if residual variance differs substantially across genes, the conversion should be read as an approximation to task-specific irreducible uncertainty rather than as a justified generative model. A natively trained NLL model—where the network outputs both $\mu_g$ and $\log \sigma_g^2$—would provide a genuinely independent estimate and is the natural next step.

## B  Detailed Run Configuration

The experiments reported in this paper use the following shared configuration unless otherwise noted:

- **Mask rate**: 15%

- **Optimizer**: AdamW with $\beta_1 = 0.9$, $\beta_2 = 0.999$, $\epsilon = 10^{-8}$

- **Weight decay**: 0.01

- **Gradient clipping**: max norm 1.0

- **Mixed precision**: float16 autocast on GPU

- **Activation**: GELU

- **Normalization**: Pre-LN (layer norm before attention and FFN)

- **Pooling**: mean pooling over gene tokens

- **Prediction head**: single linear layer

- **Initialization**: Xavier uniform for linear weights; normal ($\sigma = 0.02$) for embeddings

The standardized $V = 512$ GPU runs use physical batch size 4, gradient accumulation 8 (effective batch 32), learning rate $3.125 \times 10^{-5}$, and 60,000 steps. The fixed-$V = 512$ data-size analysis keeps the same optimization settings while varying dataset size across 25k, 50k, 100k, and 200k cells with model size M. The XXS/TINY direct-NLL reruns use the same schedule and differ only in the additional validation metric logging.

## C  Reproducibility

All code, data-building scripts, training loops, and analysis pipelines are available at [https://github.com/Biodyn-AI/scaling](https://github.com/Biodyn-AI/scaling). The OpenReview discussion page for this paper is [https://openreview.net/forum?id=a8rUQqionr](https://openreview.net/forum?id=a8rUQqionr). Datasets are constructed deterministically from the CELLxGENE Census with fixed random seeds (seed 42 for data splits, seeds 7/8/9 for model training). The full set of training run outputs—checkpoints, history logs, and metadata—is preserved in the `runs/` directory. All aggregate summaries cited in the paper were recomputed from the run manifests with explicit filtering before citation.

