# OpenReview forum: "Scaling Laws for Masked-Reconstruction Transformers on Single-Cell Transcriptomics"
_TMLR — Accepted by TMLR_

### Review · Reviewer_ACGB · 2026-03-16

**Summary Of Contributions:**

This paper studies whether masked-reconstruction transformers for single-cell RNA-seq exhibit neural scaling laws. The authors train seven model sizes on CELLxGENE-derived data under two regimes: a data-rich setting with 512 HVGs and 200,000 cells, and a data-limited setting with 1,024 HVGs and 10,000 cells. They fit a parametric scaling law L(P)=aP^(−α)+c to validation MSE and report clear scaling in the data-rich regime (α≈0.23−0.27, R^2≈0.82−0.86) but essentially no scaling in the data-limited regime (α≈0.009, R^2≈0.017). The paper further presents a preliminary entropy estimate of about 2.30 bits per masked gene position based on the asymptotic loss floor.

The main strengths are that the question is novel and relevant, the positive/negative contrast across regimes is interesting, and the paper is transparent about limitations and reproducibility. The main weaknesses are that the two regimes are confounded because both dataset size and gene vocabulary change simultaneously, Regime B includes heterogeneous training configurations that make the "no scaling" result harder to interpret, and the entropy discussion is more preliminary than the headline framing may suggest.

**Additional Comments:**

This is an interesting and timely paper, I think the strongest contribution is not the precise numerical exponent, but the broader empirical message that scaling-law behavior can emerge for masked-reconstruction scRNA-seq models in sufficiently data-rich settings, while appearing fragile in more data-constrained ones. That is a useful result for the community.

My additional suggestion is to sharpen the distinction between what the experiments directly establish and what remains an informed interpretation. With more cautious framing around the confounded regime comparison and the preliminary entropy estimate, I think the paper would be substantially stronger.

**Audience:**

Yes

**Audience Explanation:**

I expect this paper to be of interest to at least part of the TMLR audience, especially researchers working on scaling laws, self-supervised learning, biological foundation models, and single-cell representation learning. The question of whether scaling-law behavior extends to masked-reconstruction models on scRNA-seq is timely and, not well characterized in prior literature. The paper also offers a useful practical message for the single-cell foundation-model community: under the reported setup, larger models only appear to help when the training regime is sufficiently data-rich relative to the problem dimensionality. Even beyond the immediate biological application, the work is interesting as a case study in how scaling-law methodology transfers to a structured scientific domain with very different targets and noise properties than language. I therefore think at least some readers would find the findings worth knowing, even if the present version should be framed as an initial empirical study rather than a definitive scaling-law characterization.

**Broader Impact Concerns:**

The work is primarily methodological and uses aggregated single-cell transcriptomic data from a public source. That said, a brief statement could still be useful noting that foundation-model methods for biological data can inherit biases from dataset composition and preprocessing choices, and that conclusions about biological "irreducible noise" should not be overgeneralized beyond the specific preprocessing pipeline, tissue coverage, and vocabulary considered here. If the paper does not already say so, it may also be helpful to note that the reported entropy-style quantity is task- and preprocessing-dependent and should not be interpreted as a universal biological constant for transcriptomics.

**Claims And Evidence:**

Yes

**Claims Explanation:**

The paper provides reasonably clear evidence for its narrowest claim: in the 512-gene, 200k-cell regime, validation MSE follows a plausible power-law trend with parameter count, and the fit is reasonably strong for the canonical standardized subset (α=0.266, R^2=0.858). The evidence is also clear that the 10k-cell regime does not show a similarly clean trend under the reported experimental setup. The manuscript is generally well organized, the methodology is described in enough detail to follow, and the results are presented transparently.

However, the evidence doesn't fully support the strongest interpretation of the cross-regime comparison. The two regimes differ in both dataset size and vocabulary size, which the authors explicitly acknowledge as a confound. That means the paper does not cleanly isolate whether the observed difference is driven by dataset size, vocabulary size, the data-to-dimension ratio, or their interaction. In addition, Regime B includes varying training durations and even some very short runs, which weakens the claim that the flat trend directly demonstrates a fundamentally data-limited regime rather than, at least in part, inconsistent optimization budgets. Besides, the entropy claim should be interpreted cautiously. The paper itself notes that the Gaussian NLL values are derived post hoc from MSE and are therefore not an independent validation, and that a full entropy estimate would additionally require more principled loss functions and dataset-size/compute sweeps. So the evidence supports this as a preliminary exploratory estimate, but not yet as a strong quantitative conclusion.

**Requested Changes:**

1. Clarify and substantially tone down the causal interpretation of the cross-regime comparison.
Because Regime A and Regime B differ in both dataset size and gene vocabulary, the paper should avoid strong claims that the results provide direct evidence that data scarcity alone is the binding constraint. This can be presented as a plausible interpretation or hypothesis, but not as a cleanly isolated causal conclusion without additional controlled experiments.
2. Standardize or better qualify the Regime B analysis.
Since Regime B includes heterogeneous training durations and even some short runs that may not have converged, the authors should either rerun this regime under matched training budgets or clearly separate standardized runs from exploratory ones in the main analysis. As written, this heterogeneity weakens the interpretation of the near-zero scaling exponent.
3. Report test-set scaling alongside validation-set scaling.
The current analysis uses best validation MSE for fitting. Since a test split exists, it would strengthen confidence to confirm that the same qualitative conclusions hold on held-out test data.
4. Moderate claims comparing absolute MSE across regimes.
Since the two regimes differ in vocabulary size and setup, absolute MSE values are not directly comparable without stronger justification.

---

### Review · Reviewer_L3EW · 2026-03-21

**Summary Of Contributions:**

This paper studies whether there are neural scaling laws on single-cell RNA sequencing (scrna-seq) of masked-reconstruction transformers. Based on CELLxGENE Census, the author constructs two experimental regimes: a data-rich register (512 highly variable genes, 200,000 cells) and a data-limited register (1,024 genes, 10, 000 cells) are as follows: firstly, the first systematic scaling-law analysis for single cell masked-reconstruction pretraining is put forward according to the author; Secondly, clear power-law scaling and stable irreducible loss floor； are observed in Register A; Thirdly, almost disappearing scaling is observed in Register B, and the influence of data-to-parameter ratio on scaling behaviour is discussed accordingly. Fourthly, further attempt is made to convert the asymptotic floor into a preliminary entropy estimate of about 2.30 bits per masked gene position. The main advantages of this paper are that the problem incision is novel, the experimental problem is clear, and it has methodological correlation with the design of single-cell foundation models. The main weakness is that there is a confounded regime comparison in the core comparison, and the training configurations of Regime B are not unified, which makes the evidence chain of the strongest conclusion not solid enough.

**Audience:**

Yes

**Audience Explanation:**

It transplanted neural scaling laws, a mature methodology in language/vision, to single-cell transcriptomics, a rapidly developing application direction. This paper focuses not on the traditional downstream benchmark, but on the more basic Pre-training Loss Scaling, which is highly related to the researchers who pay attention to Intellectual Deep Learning, foundation models, Presentation Learning and scaling analysis in TMLR readers.

**Claims And Evidence:**

No

**Claims Explanation:**

The two regimes changed the gene vocal size (v) and dataset size (D) at the same time, and the author himself clearly admitted that this was confounding, so it is impossible to strictly isolate whether data quantity, input dimension or their joint action led to the appearance or disappearance of scaling. In addition, there are heterogeneous training configurations in Register B, including different training durations, batch sizes, and sprint experiments that may not converge, which will further weaken the persuasiveness of the causal explanation "Data Scarcity is the binding constraint".

The discussion about entropy estimate in this paper is worthy of recognition, but it should still be regarded as preliminary at present. The paper clearly shows that Gaussian NLL is derived from MSE post hoc under Gaussian assumption, rather than natively trained information-theoretical loss; At the same time, the author also acknowledges the lack of systematic L(D) sweep, L(C)/compute-optimal frontier analysis, and independently recorded Gaussian NLL.

**Requested Changes:**

Solve the unconfirmed regime comparison. At present, changing v and d at the same time is not enough to support the strong conclusion about data scarcity. At least some matched datasets should be supplemented, such as fixing V to change D and fixing D to change V, so as to distinguish data-limited from input-dimensionality effects more clearly.

Unified training protocol of Regime B. At present, different training durations, batch sizes and sprint experiments that may not converge significantly weaken the interpretability of negative results. It is suggested to use strictly standardized hyperparameters and fully convergent training settings, and re-report the results on at least one clean sweep.

---

### Review · Reviewer_4LN5 · 2026-03-25

**Summary Of Contributions:**

The paper is the first to study scaling laws for single-cell genomics. It provides empirical evidence that validation MSE of masked reconstruction transformers follows a power-law scaling with respect to model size in data-rich regimes. The paper also estimates an asymptotic MSE floor (under specific conditions) of 1.44 and attempts to convert this into information-theoretic units (entropy).

Strengths:
1) The prospect of future work, such as estimating the intrinsic entropy of the transcriptome (a more fundamental biological limit), is interesting.
2) The paper also shows that scaling laws break down in data-scarce regimes.


Weaknesses:

1) The technical rigor of the empirical experiments is lacking.
a) In Table 3, why are there two settings for A (“all runs” and “canonical runs”)? Why not use 21 (7 × 3) runs with identical hyperparameters (60k steps)? What is the purpose of having some runs with 30k steps in "all runs"?
b) Why is the number of runs in A and B not kept the same (e.g., 21 each)?


2) The mathematical justification of some derivations and statements is unclear.
a) In Section 7.4.3, the statement “approximately 2.3 bits of expression information can be predicted from the remaining genes, and the rest is irreducible noise” is vague and entirely counterintuitive. Since 2.3 bits is derived from the asymptotic MSE floor (1.44), wouldn’t it be more accurate to state that “approximately 2.3 bits of irreducible uncertainty remain per gene”?
b) The homoscedasticity assumption underlying the MSE-to-entropy conversion is not justified. Is there any reason to assume that residuals have constant variance across genes at the asymptotic floor?

**Additional Comments:**

NA

**Audience:**

Yes

**Audience Explanation:**

As noted in Strength 1, the paper should be of interest to researchers in single-cell genomics, as it helps in predicting and benchmarking large models trained on scRNA-seq data.

**Claims And Evidence:**

No

**Claims Explanation:**

The authors convincingly show that MSE follows a power-law scaling with model parameters. However, the derivation of entropy from MSE and its interpretation are not fully convincing (Weakness 2).

**Requested Changes:**

Address the weaknesses mentioned above for acceptance. In addition, would it be possible to include an empirical experiment to evaluate how close the residuals (MSE) are to being homoscedastic across genes?

---

### Author Response · Authors · 2026-04-05

We thank the reviewers for the careful and constructive feedback. We revised the manuscript substantially to sharpen the main claims, remove avoidable ambiguity, and add new analyses based on completed runs.

Main changes in the revision:

1. We removed the mixed “all runs” Regime A presentation from the main results. The main text now reports only the standardized canonical sweep (6 sizes x 3 seeds, all 60k steps with matched hyperparameters). The earlier 30k runs are now described only as exploratory pilots.

2. We added held-out test-set scaling for the canonical V=512, D=200k regime. The revised paper now reports:
- Validation: alpha = 0.2664, c = 1.4438, R^2 = 0.8580
- Test: alpha = 0.2676, c = 1.4615, R^2 = 0.8587
This confirms that the qualitative scaling conclusion is not an artifact of selecting checkpoints by validation alone.

3. We substantially toned down the cross-regime interpretation. The revised manuscript now states explicitly that the originally reported V=1024 regime is heterogeneous and exploratory, not a clean causal comparison against Regime A. We no longer present it as direct evidence that “data scarcity is the binding constraint.” Instead, we describe it as an observation consistent with a data-limited interpretation, while acknowledging that dataset size, vocabulary size, and their interaction are not isolated.

4. We removed / moderated absolute MSE comparisons across regimes, since the V=512 and V=1024 settings differ in both vocabulary size and dataset size.

5. We rewrote the entropy discussion more carefully. The paper now states that, under a Gaussian homoscedastic approximation, the fitted floor corresponds to about 2.3 bits of irreducible uncertainty remaining per masked position, not “2.3 bits of predictable expression information.” We also now state explicitly that this quantity is task-, preprocessing-, masking-, and vocabulary-dependent, not a universal biological constant.

6. We added the requested empirical homoscedasticity diagnostic. Using a representative held-out L-run checkpoint, we computed per-gene residual statistics on held-out data:
- coefficient of variation of gene-wise MSE: 0.730
- q90 / q10 ratio: 10.1
- correlation between mean gene expression and gene-wise MSE: 0.892
These results show substantial cross-gene heterogeneity, so the paper now presents the MSE-to-entropy conversion only as a task-level approximation rather than as a justified gene-wise likelihood model.

7. To partially address the D/V confounding, we added a fixed-V=512 data-size follow-up with D in {25k, 50k, 100k, 200k}, model M, 3 seeds each, all at 60k steps. At this fixed optimization budget the fit is nearly flat in MSE (beta = -0.0023, R^2 = 0.611), so we explicitly state that this does not yet provide a clean L(D) law and does not support a definitive Section 6.3-style entropy estimate.

8. We also added a limited direct Gaussian-NLL check. We completed new V=512, D=200k, 60k-step reruns for XXS and TINY with Gaussian NLL logged directly. Across 3 seeds each, mean validation Gaussian NLL differs from the MSE-derived value by about 0.02 nats for both sizes:
- XXS: 1.768 direct vs. 1.790 derived
- TINY: 1.634 direct vs. 1.654 derived
We use this only as a local calibration check of the conversion, not as a replacement for the six-size canonical table, because the available larger direct-NLL histories come from a later non-identical follow-up sweep.

In response to Reviewer 4LN5, these changes address the Table 3 ambiguity, correct the entropy wording, and add the requested homoscedasticity experiment.

In response to Reviewer L3EW, we agree that the original cross-regime comparison was confounded. We therefore weakened that claim substantially, added a fixed-V follow-up sweep, and clearly labeled Regime B as exploratory rather than as a matched negative result.

In response to Reviewer ACGB, we added held-out test scaling, moderated cross-regime MSE comparisons, added the broader-impact clarification, and sharpened the distinction between what the experiments directly establish and what remains preliminary.

What remains for the future work is a fully matched fix-D / vary-V sweep, a fully standardized V=1024 rerun, and a six-size direct-NLL canonical rerun. Because we could not complete those within the revision window, the revised manuscript now presents the cross-regime and entropy conclusions more cautiously.

---

### Decision · Action_Editor_dkNR · 2026-06-04

**Recommendation:** Accept as is

**Additional Comments:**

The reviewers all felt that the revision submitted by the authors was substantial and sufficient for publication, and I concur.

**Audience:**

Yes

**Audience Explanation:**

All reviewers were in agreement that there was a sizeable portion of readers that would find the work interesting.

**Claims And Evidence:**

Yes

**Claims Explanation:**

The reviewers were initially split on this question, with two suggesting that the scaling law findings and the entropy estimates were insufficiently well supported by the evidence, and one agreeing only in part.  However, in the revised version the claims were adjusted to the degree that the evidence would support (the addition of a held-out test set really helped here, and the standardized design of the model size sweep), and so the core new claim (that MSE on the validation set follows a power law with parameter count in standardized data-rich settings) is accepted by all reviewers.